# SPINK2 Protein Expression Is an Independent Adverse Prognostic Marker in AML and Is Potentially Implicated in the Regulation of Ferroptosis and Immune Response

**DOI:** 10.3390/ijms24119696

**Published:** 2023-06-02

**Authors:** Herbert Augustus Pitts, Chi-Keung Cheng, Joyce Sin Cheung, Murphy Ka-Hei Sun, Yuk-Lin Yung, Hoi-Yun Chan, Raymond S. M. Wong, Sze-Fai Yip, Ka-Ngai Lau, Wai Shan Wong, Radha Raghupathy, Natalie P. H. Chan, Margaret H. L. Ng

**Affiliations:** 1Blood Cancer Cytogenetics and Genomics Laboratory, Department of Anatomical & Cellular Pathology, Prince of Wales Hospital, The Chinese University of Hong Kong, Hong Kong SAR, China; 2Sir Y.K. Pao Centre for Cancer, Department of Medicine and Therapeutics, The Chinese University of Hong Kong, Hong Kong SAR, China; 3Department of Clinical Pathology, Tuen Mun Hospital, Hong Kong SAR, China; 4Pathology Department, Queen Elizabeth Hospital, Hong Kong SAR, China; 5Department of Clinical Oncology, The Chinese University of Hong Kong, Hong Kong SAR, China; 6State Key Laboratory in Oncology in South China, The Chinese University of Hong Kong, Hong Kong SAR, China

**Keywords:** acute myeloid leukemia, leukemic stem cells, prognosis, ferroptosis, immune response

## Abstract

There is an urgent need for the identification as well as clinicopathological and functional characterization of potent prognostic biomarkers and therapeutic targets in acute myeloid leukemia (AML). Using immunohistochemistry and next-generation sequencing, we investigated the protein expression as well as clinicopathological and prognostic associations of serine protease inhibitor Kazal type 2 (SPINK2) in AML and examined its potential biological functions. High SPINK2 protein expression was an independent adverse biomarker for survival and an indicator of elevated therapy resistance and relapse risk. SPINK2 expression was associated with AML with an *NPM1* mutation and an intermediate risk by cytogenetics and European LeukemiaNet (ELN) 2022 criteria. Furthermore, SPINK2 expression could refine the ELN2022prognostic stratification. Functionally, an RNA sequencing analysis uncovered a potential link of SPINK2 with ferroptosis and immune response. SPINK2 regulated the expression of certain P53 targets and ferroptosis-related genes, including *SLC7A11* and *STEAP3*, and affected cystine uptake, intracellular iron levels and sensitivity to erastin, a specific ferroptosis inducer. Furthermore, *SPINK2* inhibition consistently increased the expression of *ALCAM*, an immune response enhancer and promoter of T-cell activity. Additionally, we identified a potential small-molecule inhibitor of SPINK2, which requires further characterization. In summary, high SPINK2 protein expression was a potent adverse prognostic marker in AML and might represent a druggable target.

## 1. Introduction

Acute myeloid leukemia (AML) is an aggressive hematological malignancy that is challenging for clinical management and has a poor prognosis for patients, owing largely to its suboptimal prognostication, therapy refractoriness and high relapse risk [1,2]. However, intensive research over the past decade has contributed immensely towards enhancing our understanding of the pathobiological mechanisms underlying leukemogenesis and disease progression [3]. These findings have improved prognostic assessments in patients [4] and led to the U.S. Food and Drug Administration (FDA)’s approval of novel targeted therapies into standard clinical management for specific patient subgroups [4,5]. Nonetheless, the clinical outcome of a substantial proportion of patients remains poor [6]. Hence, the identification of potent prognostic markers and novel therapeutic vulnerabilities remains the key to ameliorating patient risk stratification and treatment [6].

Leukemic stem cells (LSCs) have been identified as potent drivers of relapse and therapy resistance, with LSC frequencies and gene expression signatures independently predicting clinical outcomes [7,8]. Furthermore, anti-LSC therapies hold great promise in substantially improving patient outcomes, since LSCs are believed to lie at the root of the disease [9,10,11]. Therefore, the clinicopathological and functional characterization of LSC-associated genes, as well as the identification and development of LSC-specific therapies, are pressing needs [12].

To this aim, we sought to identify and study LSC-associated genes that were not well characterized in AML. Our initial in silico analyses discovered serine protease inhibitor Kazal type 2 (SPINK2) to be highly expressed in functionally defined LSC fractions and suggested its specific and important roles in AML pathobiology. The prognostic significance of *SPINK2* mRNA expression has indeed been recently described in AML [13,14,15,16]. However, the clinicopathological associations of its protein expression as well as its prognostic utility in refining existing risk prediction models, predicting therapy responses and relapse risk, have not been investigated in AML. Additionally, its biological roles and therapeutic targetability in AML remain yet to be determined.

These observations and considerations provided a convincing rationale for further investigation. We discovered elevated SPINK2 protein expression by IHC to be an independent adverse prognostic biomarker with the ability to refine the ELN2022 risk assessment model. Furthermore, potential functional roles of *SPINK2* relating to ferroptosis and immune response were identified. Lastly, a putative small-molecule inhibitor (SMI) targeting SPINK2 was discovered that demonstrated desirable effects in vitro [4,17].

## 2. Results

### 2.1. Identification of SPINK2 and Assessment of Its Protein Expression in AML Patients

To screen for novel LSC-associated oncogenes, we initially analyzed several AML datasets from the Oncomine [18] and NCBI GEO [17,19] databases and identified serine protease inhibitor Kazal type 2 (*SPINK2*) with an elevated expression in AML compared to that in other leukemias, particularly in functionally defined LSC fractions (Appendix A). Next, SPINK2 expression and its clinicopathological associations in AML were determined using immunohistochemistry (IHC) and next-generation sequencing (NGS) in our cohort comprising 172 AML patients treated at the Prince of Wales Hospital (PWH). IHC for SPINK2 was performed on diagnostic BM specimens of non-M3 patients (median age: 52 years, range: 18–86 years). The majority were de novo AML (90.8%), with 72.3% having intermediate-risk (IR) cytogenetics according to the Medical Research Council (MRC) classification [20]. Appendix A summarizes their baseline characteristics. DNA was available for 152 patients and was sequenced by NGS using a targeted myeloid panel covering 141 commonly mutated genes in myeloid neoplasms. Based upon data availability, public datasets (TCGA-LAML [21], Verhaak [22], OHSU-Beat AML [23], Balgobind [24], TARGET-AML [25]) were also analyzed for clinicopathological and prognostic correlations. Details of these datasets and the exclusion criteria for the survival and treatment–response analyses are found in Appendix A. 

SPINK2’s IHC staining in leukemic blasts was consistently cytoplasmic (Figure 1A,B) and was quantified using a composite IHC score based on the percentage of stained blasts and the intensity of the staining (range: 0–16, median: 3) (Figure 1C). Furthermore, SPINK2 protein expression strongly correlated with its mRNA levels assessed by qPCR in a subset of 128 adult patients with available RNA (r = 0.716, *p* < 0.0001) (Figure 1D).

### 2.2. Mutational and Clinicopathological Associations of SPINK2 in AML

Univariate clinicopathological analyses were initially performed by dichotomization at the median SPINK2 IHC score of 3, since this cut-off exhibited the strongest association with adverse-event-free survival (EFS) and overall survival (OS) (Appendix A). SPINK2^high^ was thus defined as a score > 3 and SPINK2^low^ was defined as a score ≤ 3. SPINK2^high^ status was found in 77/172 (44.8%) patients, while SPINK2^low^ was found in 95/172 (55.2%) patients. SPINK2^high^ status associated significantly with the intermediate-risk (IR) subgroup, both by cytogenetics (*p* = 0.014) and by the European LeukemiaNet (ELN) 2022 classification (*p* = 0.009). Further significant associations were found with the normal karyotype (CN) (*p* = 0.019), *NPM1* (*p* < 0.0001) and *DNTM3A* (*p* = 0.022) mutations, including with mutational combinations, such as *NPM1+/DNMT3A+* (*p* = 0.007) and *NPM1*+/*FLT3*-ITD+ (*p* = 0.017). SPINK2^high^ status was inversely associated with t(8;21) translocation (*p* < 0.001) and *CEBPA* mutations in the basic-region leucine zipper motif (bZIP) (*p* = 0.001) (Table 1). Other commonly recurring myeloid mutations identified by NGS, including high-risk mutations such as *TP53, RUNX1* and *ASXL1*, showed no significant correlation with SPINK2 status and are listed in Table 1. Moreover, an analysis of the available cytogenetic and mutational data of 982 patients from three adult AML cohorts (TCGA-LAML, OHSU and Verhaak) largely confirmed our observations (Appendix A). 

### 2.3. High SPINK2 Expression Contributes to Therapy Resistance in AML

Survival and treatment–response analyses were initially performed on a subgroup of 137 patients that included only de novo AML patients treated on standard induction regimens with the daunorubicin and cytarabine backbone (DA 3 + 7). Complete remission (CR) was achieved by 112/137 (81.8%) patients after one or more induction courses, while 25/137 (18.2%) patients were non-responsive (NR). SPINK2^high^ patients had lower CR rates vs. those of SPINK2^low^ patients, irrespective of the number of inductions (73.3% vs. 88.3%, *p* = 0.028). Of note, non-response to first induction (NR1) was more frequent in these patients (51.7% vs. 33.8%, *p* = 0.038). Indeed, patients with NR1 had higher median *SPINK2* scores vs. those of patients with CR at first induction (CR1) (5 vs. 1.5, *p* = 0.025). 

The median relapse-free survival (RFS) of patients achieving CR was inferior in SPINK2^high^ vs. SPINK2^low^ patients (9 vs. 37 months; *p* = 0.004), with the SPINK2^high^ subgroup having more incidents of relapse within 6 months (31.8% vs. 9.1%, *p* = 0.004) (Appendix A). 

We subsequently analyzed the following subgroups due to their significant association with SPINK2 expression: IR by cytogenetics and the ELN2022, CN-AML and *NPM1^mut^* (Table 2). In most subgroups, a high SPINK2 expression was linked to lower CR rates and higher NR1 rates. Relapse risk was also elevated, achieving statistical significance in the IR groups, while demonstrating significant trends in the CN-AML and *NPM1^mut^* subgroups. The survival curves for RFS can be found in Appendix A.

The association of SPINK2 expression with outcomes after SCT was investigated next. In our cohort, 37 patients received the SCT treatment. To ascertain the association of SPINK2 and SCT outcomes, an additional 77 SCT recipients were recruited from partner hospitals, and their diagnostic BM specimens were examined for SPINK2 protein expression. In this combined transplant cohort of 114 patients (baseline characteristics: Appendix A), SPINK2^high^ status did not significantly affect OS after SCT (5 year OS: 55.8% vs. 68.8%, *p* = 0.37) (Appendix A). However, the 1 year-mortality after SCT was significantly increased in the SPINK2^high^ patients who received SCT in relapse after CR1 or in those with refractory status (61.1% vs. 5.9%, *p* = 0.041) (Appendix A).

Collectively, these findings suggest that SPINK2 might play an important role in the protection of leukemic cells against current antileukemic therapy, thereby increasing the risk of relapse. 

### 2.4. High SPINK2 Expression Refines Current Prognostic Stratification and Is an Independent Adverse Prognostic Marker

Survival analyses were initially performed on the whole cohort (N = 137), which comprised only de novo AML patients treated with the DA 3 + 7 protocol, and subsequently on specific subgroups that had significant associations with SPINK2 expression: IR risk (by cytogenetics and the ELN2022 criteria), CN-AML and *NPM1^mut^*-AML. The TCGA-LAML cohort was also analyzed. 

Univariate Kaplan–Meier analyses showed that *SPINK2^high^* status was significantly associated with inferior EFS and OS in all the aforementioned subgroups (Appendix A). 

Additionally, SPINK2 expression could identify high-risk patients among the ELN2022’s favorable-risk and intermediate-risk cohorts (Figure 2C). The incorporation of SPINK2 IHC status with the ELN2022 criteria could thus significantly refine ELN2022-based patient risk stratification (Figure 2A,B,D,E).

Importantly, multivariate analyses in our cohort highlighted the poor prognostic effect of SPINK2^high^ status on RFS (HR:1.89, 95% C.I.: 1.12–3.15, *p* = 0.015), EFS (HR:2.08, 95% C.I.: 1.31–3.32, *p* = 0.002) and OS (HR:2.45, 95% C.I.: 1.48–4.07, *p* < 0.001), independent of age, ELN2022 risk status and CR1, including SCT given in CR (Table 3). In the *NPM1^mut^* subgroup, SPINK2^high^ status predicted poor RFS (HR: 3.52, 95% C.I.: 1.23–11.72, *p* = 0.027), EFS (HR:5.11, 95% C.I.: 1.91–16.65, *p* = 0.003) and OS (HR: 5.55, 95% C.I.: 1.89–21.32, *p* = 0.005), independent of age, concomitant *FLT3*-ITD and *DNMT3A* mutational status (Table 3).

Our findings could also be observed in patients of the TCGA-LAML cohort, who had received standard DA 3 + 7-based induction regimens (N = 115). Univariate survival analyses demonstrated that higher *SPINK2* mRNA expression was associated with inferior OS in the whole cohort and in subgroups such as cytogenetic IR, CN-AML and *NPM1^mut^*. *SPINK2* expression could significantly refine risk stratification by the ELN2022’s criteria and was an independent prognostic factor for OS in the multivariate analysis (HR: 1.547, 95% C.I.: 0.99–2.43, *p* = 0.054) (Appendix A, Appendix A). 

Additionally, *SPINK2* expression remained an independent prognostic factor for OS in pairwise multivariate Cox analyses comparing *SPINK2* expression and three previously published LSC gene expression signatures (Ng [26], Gentles [27], Eppert [28]), particularly in the IR and CN subgroups (Appendix A). 

A recent study implicated *SPINK2* mRNA overexpression with primary induction failure in a large cohort of pediatric AML patients [16]. We therefore analyzed *SPINK2* mRNA expression by qPCR in our own pediatric cohort of 61 patients and found *SPINK2* mRNA overexpression to be associated with an intermediate cytogenetic risk, *FLT3*-ITD mutations, adverse survival and an elevated relapse risk (Appendix AA–D, Appendix A). Similar findings were observed in two large independent pediatric AML cohorts (Appendix AE–H, Appendix A). 

Collectively, these findings underline the prognostic importance of SPINK2 expression in AML and highlight its potential to refine current prognostic stratification by the ELN2022. 

### 2.5. Transcriptome Analysis Reveals a Potential Link between SPINK2 and Ferroptosis-Related Genes

To gain insights into the functional role of *SPINK2* in AML, its expression was initially assessed in several AML cell lines by qPCR and Western blotting, both of which showed a high expression in CD34^+^ cells (GDM1, ME-1, KG1a) and a low/negligible expression in CD34^-^ cells (NB-4, OCIAML3 and MOLM13) (Figure 3A). In KG1a cells, the *SPINK2* mRNA was knocked down (KD) with two different *SPINK2*-targeting siRNAs (#1-siRNA and #2-siRNA) using electroporation. In MOLM13 and OCIAML3 cells, *SPINK2* was overexpressed (OE) using GFP-labelled lentiviruses followed by 7-day puromycin selection. Transfection and transduction efficiency data are found in Appendix A. Differentially expressed genes (DEGs) between *SPINK2*-KD and OE cells vs. their respective negative control cells were identified by RNA sequencing (RNA-seq). SPINK2-KD with siRNA was also performed in *SPINK2*-high cells, ME1 and GDM1, for the validation of selected *SPINK2* target genes (Appendix A).

Since SPINK2 is not a transcription factor, a cut-off of 1.3 (which allowed for the incorporation of more genes for our analysis) was employed to identify commonly deregulated genes/pathways. In two independent experiments of *SPINK2*-KD in KG1a cells, 75 genes were commonly downregulated, while 99 genes were commonly upregulated by both siRNAs. In MOLM13 and OCIAML3 cells, 31 genes were commonly upregulated, while 68 genes were commonly downregulated upon *SPINK2* OE. A gene set enrichment analysis (GSEA) was performed using the Hallmark and Gene Ontology (biological processes) datasets of the Molecular Signatures Database (MSigDb) [29]. Among the top 10 enriched pathways in each dataset, the following pathways were common to both the KD and OE cells: “Interferon Gamma Response”, “Apoptosis” and “P53 pathway” (Appendix A).

Two genes were commonly upregulated in *SPINK2*-OE cells and downregulated in *SPINK2*-KD cells: *SLC7A11* and *ASNS*. *SLC7A11* is a specific cystine/glutamate antiporter and a master regulator of ferroptosis [30]. Furthermore, studies have shown that *SLC7A11* overexpression is associated with a poor prognosis in AML and that ferroptosis induction represents a novel treatment strategy [31,32,33,34,35,36]. Therefore, we investigated the relationship of *SPINK2* and *SLC7A11* further by qPCR and Western blots. Both confirmed the modulation of *SLC7A11* expression upon SPINK2-KD and OE in KG1a and MOLM13 cells (Figure 3B–D). *SPINK2*-KD in KG1a cells resulted in decreased cystine uptake and intracellular cysteine levels, which are functional consequences of *SLC7A11* downregulation (Figure 3E,F). 

Studies have shown that p53 transcriptionally represses *SLC7A11* expression, thereby playing an important pro-ferroptotic role [37,38]. Our data showed that *p53* pathway genes were inversely affected upon *SPINK2* modulation (Appendix A). We thus hypothesized that *SPINK2* overexpression in MOLM13 cells might counteract the p53-mediated repression of *SLC7A11*. MOLM13-EV and -*SPINK2* cells were treated with the p53 activator, Nutlin-3a (1 µM), for 48 h and 72 h. Indeed, the *SLC7A11* mRNA expression was reduced in the MOLM13-EV cells to a significantly greater extent than in the MOLM13-*SPINK2* cells (Figure 3G). Of note, the effects of Nutlin-3a could not be tested in KG1a cells, since they have a *TP53* mutation which renders them insensitive to Nutlin-3a activity [39].

Another notable finding was the consistent overexpression of *STEAP3* in KG1a and GDM1 cells with *SPINK2*-KD (Figure 3H). *STEAP3* is transcriptionally activated by p53 and acts as a ferrireductase (reduces intracellular ferric (Fe^3+^) to ferrous (Fe^2+^) iron) to increase the intracellular labile iron pool [40,41]. Increased intracellular Fe^2+^ is a hallmark of ferroptosis [42]. Functionally, the intracellular Fe^2+^ levels were significantly increased upon *SPINK2*-KD in the KG1a cells (Figure 3I).

Collectively, these findings suggest that SPINK2 might serve to counteract *p53*-mediated ferroptosis induction by modulating the expression of its downstream targets, *SLC7A11* and *STEAP3.*

### 2.6. Identification and Testing of SPINK2 Small-Molecule Inhibitor (SMI)

To identify potential SPINK2-SMIs, structure-based virtual screening (SBVS) was initially performed via NovoPro (NovoPro Bioscience Inc., Shanghai, China) for the in silico screening of a small-molecule library comprising 1.5 million compounds to identify bioactive molecules that bind to the targeting domain of SPINK2 (Figure 4A) [43]. Among the top one thousand compounds with a higher affinity to SPINK2 based upon their idock scores [44], we filtered out the top four that were purchasable. Only one of these compounds (C_26_H_19_NO_4_, PubChem CID: 1102833) (Figure 4B) was soluble in DMSO and was therefore chosen for the further analysis. The other three compounds were insoluble in DMSO and other available solvents, such as water, ethanol and dimethylformamide. 

The SMI was initially tested with increasing doses for its effect upon cell viability in KG1a cells at different time intervals. At 72 h, the 150 µM treatment reduced cell viability by approximately 50% (Figure 4C). This dose was then tested on GDM1, MOLM13 and OCIAML3 cells. The SMI treatment (150 µM for 72 h) significantly decreased the cell viability of *SPINK2^high^* cells (KG1a, GDM1) but not *SPINK2^low^* cells (OCIAML3, MOLM13) (Figure 4D). SPINK2 protein expression was also decreased in KG1a and GDM1 cells by the SMI treatment (Figure 4E). Additionally, the SMI treatment of KG1a cells resulted in an alteration in SPINK2 target gene mRNA expression, namely the downregulation of *SLC7A11* and the upregulation of *STEAP3* (Figure 4F)—consistent with the effects observed via genetic *SPINK2* inhibition with siRNAs. 

### 2.7. Genetic and Pharmacologic Modulation of SPINK2 Expression Influences Sensitivity to Erastin, a Ferroptosis Inducer

The identified link between *SPINK2*, *SLC7A11* and *STEAP3* led us to hypothesize that *SPINK2* inhibition might render the cells more susceptible to ferroptosis induction since both *SLC7A11* and *STEAP3* perform key functions in ferroptosis regulation [30,45]. Erastin is a small molecule that potently and specifically inhibits SLC7A11, resulting in ferroptotic cell death [46]. To test our hypothesis, we examined the effects of *SPINK2* modulation upon erastin sensitivity. 

Forty-eight hours after *SPINK2*-KD, KG1a cells were treated with a range of erastin doses (2.5–10 µM) for 24–48 h. The cell viability was significantly reduced in the *SPINK2*-KD cells vs. that in the negative control upon the erastin treatment (Figure 5A). Next, MOLM13-*SPINK2* and -EV cells were similarly treated with erastin for 48–96 h. MOLM13-*SPINK2* cells were significantly more resistant to cell death by erastin at 96 h (Figure 5B). The effects of pharmacologic *SPINK2* inhibition with the potential SPINK2-SMI on erastin were also examined. Wildtype KG1a and GDM1 cells were treated with a combination of erastin (2.5 µM) and/or SPINK2-SMI (150 µM) for 72 h. The combined erastin/SMI treatment significantly reduced cell viability compared to that of using erastin alone (Figure 5C). Interestingly, cell death with the apoptosis inducer cytarabine (Ara-C) was not significantly altered in KG1a cells with *SPINK2*-KD (Figure 5D). However, there was mild reduction in cell viability in the MOLM13-SPINK2 cells treated with Ara-C (Figure 5E). Collectively, these findings suggest that *SPINK2* might be involved in molecular pathways that mediate ferroptotic rather than apoptotic cell death. 

### 2.8. SPINK2 Modulation Affects Expression of Immune-Response-Related Genes in LSC-Like Cells

Avoiding destruction by the immune system is one of the several hallmarks of cancer cells [47]. Immune evasion is indeed a prominent characteristic of AML blasts and LSCs [48,49]. Our analysis uncovered a potential link between *SPINK2* and immune response regulation. Among the DEGs in *SPINK2*-KD KG1a cells, the expression of several immune-response-related genes was strongly altered (≥two-fold). Among the upregulated genes was activated leukocyte cell adhesion molecule (*ALCAM*), a potent T-cell activator [50]. Interestingly, ALCAM expression was consistently increased in the LSC-like KG1a, ME1 and GDM1 cells with *SPINK2*-KD (Figure 6A,B). Additionally, T-cell activity inhibitory genes (*CD86, S100A9, NQO1*) were downregulated in the KG1a cells. This was validated by qPCR in three independent knockdown experiments (Figure 6C, blue arrows). The GSEA analysis indeed showed that the pathways involved in immune system regulation were affected in the KG1a cells (Appendix A). Collectively, these findings suggest that SPINK2 might contribute to immune evasion by the suppression of T-cell activity in LSCs. 

## 3. Discussion

The present study reports the results of a detailed clinicopathological investigation and functional assessment of an LSC-associated gene, *SPINK2,* in AML. A few studies have indeed reported *SPINK2* mRNA overexpression in conjunction with poor prognoses in AML—either as a single gene or in combination with other genes [13,14,15,51,52,53]. Nevertheless, in-depth analyses of its protein expression, clinicopathological associations and prognostic utility in predicting therapy responses in AML are lacking. Furthermore, importantly, the functional role and therapeutic targetability of SPINK2 in AML remain yet to be determined. 

To the best of our knowledge, our study is the first to analyze SPINK2 protein expression in a large adult AML cohort using IHC and thus to determine its clinicopathological associations and prognostic utility. SPINK2 protein expression was found to represent a potent prognostic biomarker which could further refine risk stratification defined by the latest ELN 2022 criteria and additionally reliably predict therapy outcomes. 

Our initial in silico analyses of several public AML datasets demonstrated high levels of *SPINK2* mRNA expression in AML compared with normal bone marrow, particularly in functionally defined LSC fractions. Though several members of the *SPINK* gene family, particularly *SPINK1*, have been associated with aggressive cancer phenotypes, little is known about *SPINK2* in cancer and AML [54]. Initial reports have suggested its important oncogenic role in the development of lymphomas and leukemias [55,56]. On the contrary, a tumor-suppressive role involving the inhibition of the epithelial–mesenchymal transition was also recently described for *SPINK2* in testicular cancer [57]. In normal tissues, high *SPINK2* expression has been detected in the testis and found to be crucial for normal sperm development as an acrosin inhibitor [58]. Interestingly, the most primitive hematopoietic cells also possess markedly high levels of *SPINK2*, suggestive of its potential role in stemness maintenance [59,60]. 

We detected high SPINK2 protein expression in intermediate-risk, normal karyotype and *NPM1^mut^* subgroups. Among these subgroups, the SPINK2 expression could identify high-risk patients. Notably, these genetic categories constitute large proportions of AML patients with a high degree of clinical heterogeneity, in need of potent biomarkers to refine prognostication and guide therapy decisions [61,62]. The prognostic effect of SPINK2 in the whole cohort was independent of potent markers, such as age, cytogenetics, ELN2022 risk status, and CR at first induction. SPINK2 status could also refine risk stratification by the ELN 2022’s criteria—identifying higher risk patients among those classified as favorable or intermediate. Additionally, no significant correlations were detected between SPINK2 expression and known high-risk mutations in *RUNX1, ASXL1, TP53* and other myelodysplasia-related genes. A possible explanation might be that our study cohort was relatively young (median age: 52 years) and contained largely de novo AML patients, where the occurrence of these high-risk mutations is relatively lower as compared with older AML patients and those with antecedent malignancies [63]. Nonetheless, our analysis of two independent publicly available datasets (TCGA-LAML and OHSU-BEAT-AML) also failed to demonstrate any significant association between *SPINK2* mRNA expression and these high-risk mutations. Thus, SPINK2 protein expression might indeed have important an added prognostic value in AML. 

SPINK2 was also linked to therapy resistance and increased relapse rates in adult AML patients. High SPINK2 expression was associated with resistance to standard induction using daunorubicin and cytarabine and was an independent marker for relapse. Patients with a SPINK2^high^ status were at a higher risk of early relapse after achieving CR. Thus, SPINK2 might represent a marker of MRD, and further investigation into its utility for the improvement in MRD detection is warranted. An assessment of diagnostic SPINK2 status might thus be clinically beneficial to screen for patients who would require more frequent monitoring for early relapse detection and treatment and those who might benefit from upfront alternate treatment strategies.

Currently, venetoclax combined with azacitidine is an alternative treatment strategy approved for AML treatment and is effective in eradicating LSCs [4,64]. However, a substantial proportion of venetoclax-treated patients are refractory, and a further subset displays MRD positivity post-remission [65]. The association between venetoclax sensitivity and SPINK2 expression is still unknown and might be an interesting point for further investigation. Indeed, the identification of patient subgroups who would most benefit from venetoclax-based regimens is a pressing clinical need [11].

SPINK2 status also identified patients who might not benefit from SCT given as salvage therapy. It is known that pre-transplant MRD levels are an important determinant of transplant outcomes and predict poor survival post-transplant [66,67]. It is possible that high diagnostic SPINK2 levels are reflective of a greater MRD burden and leukemic cell aggressiveness in patients at relapse or in those with refractory status. This might have mitigated the curative effects of the SCT given as salvage. However, these findings must be interpreted with caution, since neither the pre-transplant SPINK2 levels nor MRD burden were known. Further investigation is required in larger studies with a determination of the SPINK2 levels at diagnosis and relapse.

Our preliminary functional assessment in AML cell lines revealed novel potential functional roles of *SPINK2*, namely in the regulation of ferroptosis and immune response. Ferroptosis is a morphologically distinct form of programmed cell death that involves the iron-dependent lipid peroxidation of cell membranes [17]. Since its discovery a decade ago, ferroptosis has attracted great attention in the scientific community, and numerous studies have demonstrated its involvement in various pathophysiological (cancer, infection, autoimmune diseases) and physiological processes [68]. Ferroptosis induction represents a novel and promising therapeutic vulnerability in cancer, as well as in eliminating cancer stem cells [69,70]. 

One of the primary cellular anti-ferroptotic defense mechanisms involves the SLC7A11-GPX4-GSH axis [30,69]. *SLC7A11* associates with *SLC3A2* to form the xCT complex, which imports cystine into the cells and is considered the major source of intracellular cysteine and glutathione [30,71]. *SLC7A11* likely plays an important role in LSC biology, since its overexpression has been linked to poor prognoses in AML, and LSCs are critically reliant on cysteine for the sustenance of their energy metabolism [31,32,72]. Anti-ferroptotic defence mechanisms might thus represent a crucial survival strategy in AML cells, since ferroptosis induction has been found to increase their sensitivity to chemotherapy [34,35,36]. Our transcriptomic analysis uncovered a link between *SPINK2* and *SLC7A11*. The modulation of *SPINK2* expression affected *SLC7A11* expression and resulted in functional consequences attributable to *SLC7A11*, such as cystine uptake and an altered sensitivity to erastin, a ferroptosis inducer. Our data also suggest that *SPINK2* might be involved in the suppression of *p53*-mediated ferroptosis induction. The tumor-suppressor *p53* is now a well-known master regulator of ferroptosis and a transcriptional repressor of *SLC7A11* [38]. The expression of another *p53* target, *STEAP3*, which is pro-ferroptotic and increases intracellular Fe^2+^ [40,41], was also affected by *SPINK2* modulation with resultant functional changes (i.e., increased Fe^2+^ levels). These findings suggest that SPINK2 might play a role in pathways relating to ferroptosis by acting downstream of p53 to inhibit its effects and thus promote cell survival. 

Although these observations suggested the role of *SPINK2* in ferroptosis-related pathways, definitive evidence of its involvement in ferroptosis regulation in AML is necessary by performing ferroptosis-specific analyses, such as a determination of lipid peroxidation levels and mitochondria morphological changes, as well as an assessment of cell death reversal by ferrostatin-1, a specific ferroptosis inhibitor [17]. These issues would need to be addressed in future studies.

Evading the immune system is a hallmark of cancer and an important survival mechanism employed by AML blasts and LSCs [47,48]. Of note, an analysis of insilico data in a recent study discovered a link between *SPINK2* and immune regulation via PI3K-AKT signalling and PD-L1 expression [13]. Our study provides functional evidence showing that *SPINK2* regulates the expression of immune-response-related genes, particularly in LSC-like cells. 

*SPINK2* knockdown consistently increased the expression of *ALCAM* in three LSC-like cell lines: KG1a, ME1 and GDM1. *ALCAM*, an immunoglobulin superfamily protein, is expressed by antigen-presenting cells (APCs) and is a specific ligand of the CD6 receptor on CD4^+^ T-cells [73]. The *CD6/ALCAM* interaction is crucial for the establishment of the immunological synapse, which promotes T-cell activation and proliferation [50,74,75,76]. The GSEA analysis of our RNA-seq data further showed that several pathways associated with the regulation of immune response were affected by *SPINK2* knockdown and overexpression. *SPINK2* might thus serve to mitigate the immune response by modulating the expression of genes associated with T-cell activity, especially *ALCAM* expression. *SPINK2* is normally highly expressed in the testis, where it is essential for normal spermiogenesis and where the spermatozoa must be protected from eradication by the immune system [58,77]. High *SPINK2* expression in LSCs might help boost their survival against the host immune system. 

Interestingly, recent studies have also demonstrated a link between anti-tumor immune response and ferroptosis [78,79]. Wang et al. discovered that activated CD8^+^ T-cells induced ferroptotic cell death in cancer cells by downregulating *SLC7A11* expression through interferon-gamma secretion [79]. Given the link we identified between *SPINK2*, ferroptosis and immune response, it would be worthwhile to further study the functions of *SPINK2* in this context in an in vivo model.

Finally, we have identified a potential *SPINK2* small-molecule inhibitor (SMI) that selectively decreased the viability of high-*SPINK2*-expressing cells (KG1a, GDM1), decreased *SPINK2* protein expression, altered the expression of *SPINK2* targets (*SLC7A11* and *STEAP3)* and increased erastin sensitivity. To the best of our knowledge, our study is the first to identify and provide an initial in vitro validation of a specific SPINK2-SMI. This SMI might prove to become a specific anti-LSC therapy since it targets a specific LSC-associated oncogene. Further functional characterization is needed to determine its therapeutic potential, which might pave the way for a novel treatment approach to target residual LSCs, thereby improving the outcomes for this aggressive malignancy. 

In conclusion, our study provides strong clinical evidence of SPINK2 protein expression as a potent biomarker in AML. SPINK2 expression could be used to refine prognostic stratification according to the ELN2022’s criteria and was an indicator of adverse clinical outcomes, elevated relapse risk and therapy resistance. Functionally, SPINK2 might be involved in protecting leukemic cells from cell death by ferroptosis and enhancing their immunoevasive ability. Further studies are needed to validate our clinicopathological findings and explore in depth the functions and therapeutic targetability of SPINK2 in AML.

## 4. Materials and Methods

### 4.1. Antibodies and Drugs

The following primary antibodies were used: SPINK2 (#HPA026813, Atlas Antibodies, Stockholm, Sweden), SLC7A11 (#12691S, Cell Signaling Technology, Danvers, MA, USA), ALCAM (#ab109215, Abcam, Boston, WA, USA), β-Actin (#ab8266, Abcam) and GAPDH (#ab9485, Abcam).

The following drugs were used: Dimethyl sulfoxide (DMSO, #D4540, Sigma-Aldrich, Burlington, MA, USA), Nutlin-3a (#S8059, Selleckchem, Houston, TX, USA), erastin (#5499, Tocris, Bristol, UK), Puromycin (#A1113802, ThermoFisher, Waltham, MA, USA) and C_26_H_19_NO_4_ (#OSSK_987997, Princeton Biomolecular Research, Monmouth Junction, NJ, USA). The drugs were used at the concentrations indicated in the main text. 

### 4.2. Immunohistochemistry (IHC)

IHC for SPINK2 expression was performed on the fully automated Ventana Benchmark ULTRA platform (Ventana Medical Systems, Inc., Tucson, AZ, USA). Specimens were sectioned at a thickness of 4 µm, stained on positively charged glass slides and stored at room temperature until further use. Initially, the slides were warmed at 70 °C for 10–15 min. Deparaffinization, rehydration and antigen retrieval were performed on the Ventana’s automated slide stainer using CC1 antigen retrieval solution (Ventana Medical Systems, Inc.) at 100 °C for 64 min. Incubation with primary SPINK2 antibody was performed at a dilution of 1:100 for 32 min at 36 °C. The OptiView DAB (3,3′-Diaminobenzidine) IHC Detection Kit v5 (Ventana Medical Systems, Inc.) was then used for visualization, involving post-primary peroxidase blocking for 4 min and incubation with Linker and Multimer solutions (Ventana Medical Systems, Inc.) for 12 min each. Slides were then incubated with hydrogen peroxide and DAB for 8 min, followed by copper enhancement for 4 min. Next, counterstaining was performed with Mayer’s Haematoxylin for 1–2 min, followed by bluing agent for 1 min and standard manual dehydration with ethanol and xylene. Slides were coverslipped and warmed for 10 min prior to microscopic analysis. Normal testicular tissue served as a positive control (with buffer and primary antibody) and negative control (with buffer, without primary antibody). Slide images were captured using Nikon Ni-u Light Microscope (Nikon Instruments Inc., Melville, NY, USA). 

### 4.3. In-House Adult AML Patient Dataset and Exclusion Criteria for Survival Analysis

A total of 172 non-M3 adult AML patients treated at the Prince of Wales Hospital (PWH) in Hong Kong were recruited into the study. Archival formalin-fixed paraffin-embedded diagnostic bone marrow trephine biopsies or clots were analyzed for *SPINK2* protein expression by immunohistochemistry (IHC) using the fully automated Ventana BenchMark ULTRA platform (Ventana Medical Systems, Inc., Tucson, AZ, USA). Thirty-five patients were excluded from the survival and treatment-response analyses because of the following reasons: (i) secondary or therapy-related AML, or AML with myelodysplasia-related changes (n = 10); (ii) not receiving standard induction therapy with the daunorubicin–cytarabine (DA) 3 + 7 backbone (n = 14); (iii) loss of clinical follow up (n = 5); and (iv) death within days of diagnosis or induction (n = 6). Thus, for more accurate and non-biased survival and treatment–response analyses, a relatively homogeneous cohort of 137 de novo AML patients receiving standard DA 3 + 7 backbone regimens at induction was studied. Forty-one patients received SCT, of which only thirty-seven were included in the survival and treatment response analysis based upon the exclusion criteria mentioned above. To examine the association of *SPINK2* status and SCT outcome, an additional 77 SCT recipients with de novo AML who were receiving DA 3 + 7 induction therapy backbone were recruited from partner hospitals to generate a combined SCT cohort (N = 114). Of these, 82 (71.9%) patients received SCT at CR1, while the remainder received SCT as salvage—either in relapse or with primary refractory status. Data collection for clinical information was ended in March 2021.

### 4.4. Definition of Clinical End-Points

Overall survival (OS) was defined as the time from date of diagnosis until date of last follow-up or death by any cause. Event-free survival (EFS) was defined as time elapsed from date of diagnosis until date of first leukemic event (non-response to therapy, relapse or death) or last follow-up. Relapse-free survival (RFS) was defined as time elapsed from date of achievement of complete remission (CR) until date of relapse or death (from any cause) or last follow-up. For the transplant analysis, post-SCT OS was defined as the time elapsed from receipt of SCT until death from any cause or last clinical follow-up. CR was defined according to standard criteria [4].

### 4.5. SPINK2 IHC Score Calculation and Prognostic Cut-Off Determination

SPINK2 IHC expression was assessed independently by 3 qualified hematopathologists (J.S.C., M.K.H.S, and M.H.L.N), blinded to each other and to the clinical data of the patients. Quantification of SPINK2 expression was achieved through a composite SPINK2 IHC score employing the percentage of stained blasts (P) and the intensity of the staining (I). ‘P’ values were as follows: <20% = 1, 20–50% = 2, 50–75% =3, >75% = 4. ‘I’ values were as follows: negative—0, mild—1, moderate—2, strong—3, very strong—4. Each patient received a unique score calculated as ‘P × I’. The minimum score was 0, the maximum 16. The average of the pathologists’ scores was assigned as the final score for each patient. 

In order to determine an optimal expression cut-off with strongest prognostic implications, the cohort of 137 patients was initially divided into 4 quartiles (q1, q2, q3 and q4) based upon SPINK2 score distribution (q1: score 0, q2: score 1–3, q3: score 4–7, q4: score 8–16). Kaplan–Meier univariate survival analyses for OS and EFS showed that dichotomizing patients by the median score of ‘3′ had the strongest association with adverse outcome in terms of the log-rank *p*-value and hazard ratio (HR) when each quartile was compared with the others (Appendix A).

### 4.6. RNA Extraction, Quantitative Polymerase Chain Reaction (qPCR)

Total RNA was extracted using the QIAamp RNA Blood Mini Kit (Qiagen, Hilden, Germany). cDNA was synthesized using the Superscript III First-Strand Synthesis System (ThermoFisher) according to the manufacturer’s instructions. qPCR was performed using the Applied Biosystems 7300 real-time PCR system (Applied Biosystems). The following conditions were employed: Hold (50 °C, 2 min)—Hold (95 °C—10 min)—40 cycles (95 °C, 15 s—60 °C, 1 min). The following TaqMan^®^ Gene Expression Assay was used for SPINK2: Hs01598293_m1. Each sample was measured in triplicate, and gene expression was analyzed by the 2^−∆∆Ct^ method. GAPDH was used as housekeeping gene for normalization. The relative fold change in SPINK2 in clinical samples was compared to the expression in sorted CD34^+^ cord blood cells. RNA was available for 128 patients, and SPINK2 mRNA levels were assessed by qPCR for correlation analysis with IHC scores in these patients. 

### 4.7. Targeted Next-Generation DNA Sequencing

In most cases, diagnostic BM was used for genomic DNA extraction with Gentra Puregene Blood Kit (Qiagen, Hilden Germany). In some cases, genomic DNA was extracted from diagnostic peripheral blood (PB). Details are found in Appendix A. DNA concentration was determined with the Qubit^TM^ dsDNA BR Assay Kit (ThermoFisher, Waltham, MA, USA). Libraries were prepared following the manufacturer’s protocol from 10 ng of genomic DNA using the unique molecular identifier (UMI)-based QIAseq Targeted Human Myeloid Neoplasms Panel (Qiagen, cat# DHS-003Z), which encompasses the exon region of 141 myeloid-related genes (Appendix A). Purified and amplified libraries were then sequenced on an Illumina NextSeq 550 system. The UMI-based variant caller smCounter2 was then used on GeneGlobe (Qiagen) to analyze the sequencing data, which included read processing, alignment (version hg19) and calling of single nucleotide variants (SNVs)/small indels [80]. Variant annotation was performed by ANNOVAR [81]. Variant filtering was performed to a large extent according to the multi-step method previously described by the German AML Cooperative Group [82]. Initially, a variant allele frequency (VAF) of 5% with a quality score of 15 was chosen as cut-off for variant filtering. Synonymous SNVs were also removed, while non-synonymous, frameshift, splicing-site mutations were considered pathogenic and retained. Additionally, variants reported in OncoKB [83] as pathogenic/likely pathogenic, oncogenic/likely oncogenic or known drivers were kept. Secondly, variants with a population frequency of ≥0.1% in the 1000 Genomes Project (Phase 3) were excluded from the analysis. Finally, variants that had a Combined Annotation Dependent Depletion (CADD) [84] score >20 and that were predicted to be functionally damaging by at least 3 of the following prediction tools were retained: SIFT [85], Polyphen_2 [86], MutationTaster [87] and PROVEAN [88]. The final list of high-confidence variants is found in Appendix A. In addition, Genetic Analyzer 3500 (ThermoFisher) was also used to screen for NPM1, FLT3-ITD and CEBPA mutations. For NPM1, screening involved C-terminus mutations in exon 12, and the mutation type was reported according to pre-defined criteria [89]. All patients were screened for FLT3-ITDs using fragment analysis and Sanger sequencing. CEBPA genotyping was performed using conventional Sanger sequencing. 

### 4.8. Cell Lines and Cell Culture

GDM-1, KG1a, ME-1, K562 and NB-4 cells were obtained from American Type Culture Collection (ATCC) or German Collection of Microorganisms and Cell Cultures GmbH (DSMZ). MV4–11 and MOLM13 were kindly provided by Prof. Kam Tong Leung (Department of Paediatrics, The Chinese University of Hong Kong, Hong Kong SAR, China). OCI-AML3 cells were kindly provided by Prof. M.D. Minden (Princess Margaret Cancer Centre, University Health Network, Toronto, ON, Canada). ME-1 and GDM1 cells were maintained in RPMI-1640 medium containing 20% fetal bovine serum (FBS), while all others were maintained in RPMI-1640 medium with 10% FBS. 

### 4.9. RNA Interference

For SPINK2 knockdown, predesigned siRNAs were purchased from Ambion (siRNA#1: ID_s13362, siRNA#2: ID_s224675). Negative control siRNAs were also obtained from Ambion (Cat #AM4611). Approximately 5 × 10^6^ cells in RPMI1640 medium were transfected with 500 nM siRNAs using electroporation (Bio-Rad Gene Pulser Xcell^TM^) with 0.4 cm cuvettes and the following conditions: voltage, 300 V; and capacitance, 700 µF. Forty-eight to seventy-two hours after transfection, SPINK2 expression was analyzed by qPCR and Western blotting. 

### 4.10. Lentiviral Transduction

GFP-labelled lentiviruses for SPINK2 (pRSC-SFFV-SPINK2-E2A-Puro-E2A-GFP-Wpre) and empty vector, EV (pRSC-SFFV-Puro-E2A-GFP-wpre), were kindly provided by Prof. Kam Tong Leung (Department of Paediatrics, The Chinese University of Hong Kong, Hong Kong SAR, China) in liaison with Prof. Xiao-Bing Zhang (Department of Medicine, Loma Linda University, Loma Linda, CA, USA). Transduction was performed in approximately 2 × 10^5^ cells/mL at a multiplicity of infection (MOI) of 20 using Retronectin^®^-coated 6-well plates according to the manufacturer’s instructions (Takara Bio Inc., Kusatsu, China). This was followed by puromycin selection (1 µg/mL) for at least 7 days. Functional studies were then performed on cells as described, and extra cells were cryopreserved. 

### 4.11. Transcriptome Sequencing

Transcriptome sequencing was performed to identify gene expression changes upon SPINK2 knockdown (KD) and overexpression (OE). Total RNA was extracted from two independent experiments involving KG1a cells transfected with negative-control siRNA, SPINK2 siRNA#1 and SPINK2 siRNA#2 for 48 h. Total RNA was also extracted from MOLM13 and OCIAML3 cells transduced with EV and SPINK2 lentiviruses, following a 7-day puromycin selection period. All the subsequent steps involving mRNA purification from total RNA, library preparation, sequencing on the Illumina NovaSeq 6000 system and data analysis (quality control, reference genome mapping (version hg19) and quantification of gene expression level) were performed by Novogene (Novogene Co., Ltd., Beijing, China). For quantification of gene expression levels, FPKM (fragments per kilobase of transcript per million mapped reads) of each gene was calculated based on the length of the gene and read count mapped to it. We then performed differential gene expression analysis manually by excluding non-protein coding genes and those with FPKM < 1 in the control cells. Next, the FPKM of genes of the KD or OE cells was divided by the FPKM of genes of the control cells to generate the fold change for each gene. A fold change of 1.3 was chosen as a cut-off for both downregulation and upregulation analysis to incorporate more genes for Gene Set Enrichment Analysis (GSEA), since SPINK2 is not a transcription factor. 

Quantitative RT-PCR was employed to validate selected SPINK2 target genes using the following TaqMan Gene expression assays: SLC7A11 (Hs00921938_m1), STEAP3 (Hs00217292_m1), ALCAM (Hs00977641_m1), CD86 (Hs01567026_m1), NQO1 (Hs01045993_g1), S100A9 (Hs00610058_m1), VWF (Hs01109446_m1), ITGA2B (Hs01116228_m1), IL32 (Hs00992441_m1), CCNA1 (Hs00171105_m1), HOXA6 (Hs00430615_m1), TFPI (Hs00409210_m1), CDH24 (Hs00332067_m1) and MDK (Hs00171064_m1). Each sample was measured in triplicate and gene expression was analyzed by the 2^−∆∆Ct^ method. GAPDH was used as housekeeping gene for normalization.

### 4.12. Biochemical Assays

Cystine uptake level was measured in KG1a cells using the Cystine Uptake Assay Kit from Dojindo Laboratories (#UP05) according to the manufacturer’s instructions. Cystine-free Dulbecco’s Modified Eagle Medium (DMEM) was purchased from ThermoFisher (Cat #21013024) for use with the assay. Intracellular cysteine and iron (Fe^2+^) levels were quantified in KG1a cells using the fluorometric Cysteine Assay Kit (#ab211099, Abcam) and colorimetric Iron Assay Kit (#ab83366, Abcam) following the manufacturer’s instructions. 

### 4.13. Western Blotting

Cells were harvested, washed in phosphate-buffered saline (PBS) and lysed using Pierce^TM^ IP Lysis Buffer (ThermoFisher, #87787) according to the manufacturer’s instructions. Protein concentration was measured using Pierce^TM^ BCA Protein Assay Kit (ThermoFisher, #23225). Approximately 30 µg of whole cell lysates was mixed with 4× Laemmli Buffer (#1610747, Bio-Rad, Hercules, CA, USA) and β-mercaptoethanol (#1610710, Bio-Rad) and denatured for 5 min at 95 °C. Lysates were equally loaded onto and separated using freshly prepared polyacrylamide gels. Proteins were transferred onto 0.2 µm Immun-Blot^®^ PVDF membranes (Bio-Rad, #1620174) using FLASHBlot transfer buffer (#R-03090-D50, Advansta, Inc., San Jose, CA, USA). The membranes were then blocked for 1 h at room temperature with 5% non-fat dry milk (Cell Signaling Technology, Danvers, MA, USA, #9999) in TBS Tween^TM^ 20 Buffer (#28360, ThermoFisher). This was followed by incubation with primary antibodies diluted in 5% bovine serum albumin (BSA) at 4 °C overnight. Primary antibody dilutions were as follows: SPINK2 (1:1000), ALCAM (1:10,000), β-ACTIN (1:10000) and GAPDH (1:2500). Membranes were washed with 1× TBS Tween™ and incubated for 1 h at room temperature with species-specific horseradish peroxidase-labelled (HRP) secondary antibodies—either goat anti-rabbit IgG-HRP (#P0448, Dako/Agilent, Santa Clara, CA, USA,) or goat anti-mouse IgG-HRP (#P0447, Dako/Agilent, Santa Clara, CA, USA), both at 1:2000 in 5% BSA. Chemiluminescent detection was then performed after incubation of the membranes with WesternBright ECL HRP Substrate (Advansta, Inc.) and imaging using the ChemiDoc XRS+ System (Bio-Rad).

### 4.14. Drug Treatment and Cell Viability Assays

Cells were seeded into 96-well plates at a density of approximately 2 × 10^5^ cells/mL, and drugs were added at indicated concentrations. Cell viability was measured at indicated time points using Cell Titer-GLO^®^ Luminescent Cell Viability Assay (Promega, Madison, WI, USA) according to the manufacturer’s protocol. For assessment of gene expression after drug treatment, cells were seeded in 6-well plates at approximately 4 × 10^5^ cells/mL, and drugs were added at indicated doses. RNA and/or protein was extracted 72 h later.

### 4.15. Statistical Analyses

All statistical analyses were performed using GraphPad Prism version 9 for Windows (GraphPad Software, San Diego, CA, USA). Various two-tailed *t* tests were used for comparison of clinicopathological characteristics between patients with SPINK2^high^ and SPINK2^low^ status: unpaired Student’s *t*-test, Mann–Whitney test or Kruskal–Wallis tests were used for continuous variables, whereas Fisher’s exact test was used for categorical variables. For comparison of response to standard induction among SPINK2^high^ and SPINK2^low^ groups, Fisher’s exact test was used. For univariate survival analyses, Kaplan–Meier curves were generated, and the logrank *p*-value and logrank hazard ratio were used for comparison of groups. *p*-values < 0.05 were considered statistically significant. For multivariate analysis, we first performed univariate survival analysis with Cox regression for several variables and/or combinations individually. Factors which were significantly associated with survival in the univariate analysis were then input into the multivariate analysis. In the multivariate analysis results, *p*-values < 0.05 were considered statistically significant. For all other tests in the functional assays, the statistical test employed is indicated in the main text. The data are presented for at least 2 independent experiments as mean ± standard deviation (SD) as indicated in figure legends.

## Figures and Tables

**Figure 1 ijms-24-09696-f001:**
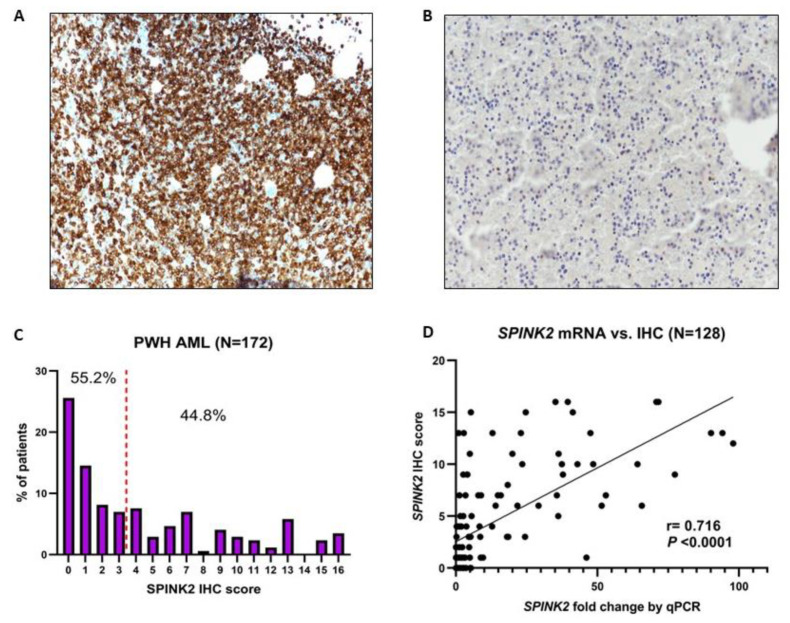
SPINK2 IHC staining and expression in adult AML. (**A**,**B**) Representative IHC images with SPINK2 staining with a dilution of 1:100 showing strong (**A**) expression and low/absent (**B**) expression. Images were captured using a Nikon Eclipse Ni DS-Ri2 Microscope (Nikon Instruments, Inc.) at 20× magnification using the NIS software. (**C**) Histogram showing the SPINK2 IHC score distribution among 172 adult AML patients of the PWH cohort. The height of the purple bars denotes the percentage of patients having a particular score from 0 to 16. A score of 3 was the median (red dashed vertical line). (**D**) Strong positive correlation between SPINK2 IHC score and mRNA fold change by qPCR in a subset of 128 patients.

**Figure 2 ijms-24-09696-f002:**
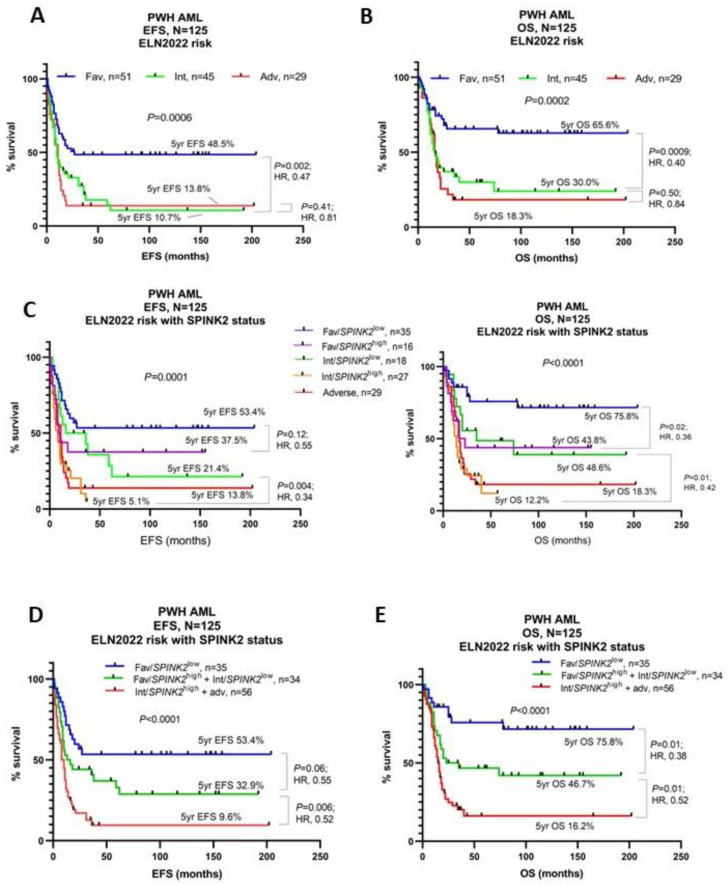
Prognostic refinement of ELN2022 risk with SPINK2 IHC status. (**A**,**B**) Kaplan–Meier (KM) survival curves for EFS (**A**) and OS (**B**) based upon ELN2022 risk only. (**C**) KM curves for EFS (**left**) and OS (**right**) based upon ELN2022 risk with incorporation of SPINK2 IHC status. (**D**,**E**) KM curves for EFS (**D**) and OS (**E**) based upon ELN2022 risk with incorporation of SPINK2 IHC status and combination of indicated categories. Survival proportions were compared using the logrank *p*-value and logrank hazard ratio (HR). Abbreviations: Fav, favorable risk; Int, intermediate risk; Adv, adverse risk.

**Figure 3 ijms-24-09696-f003:**
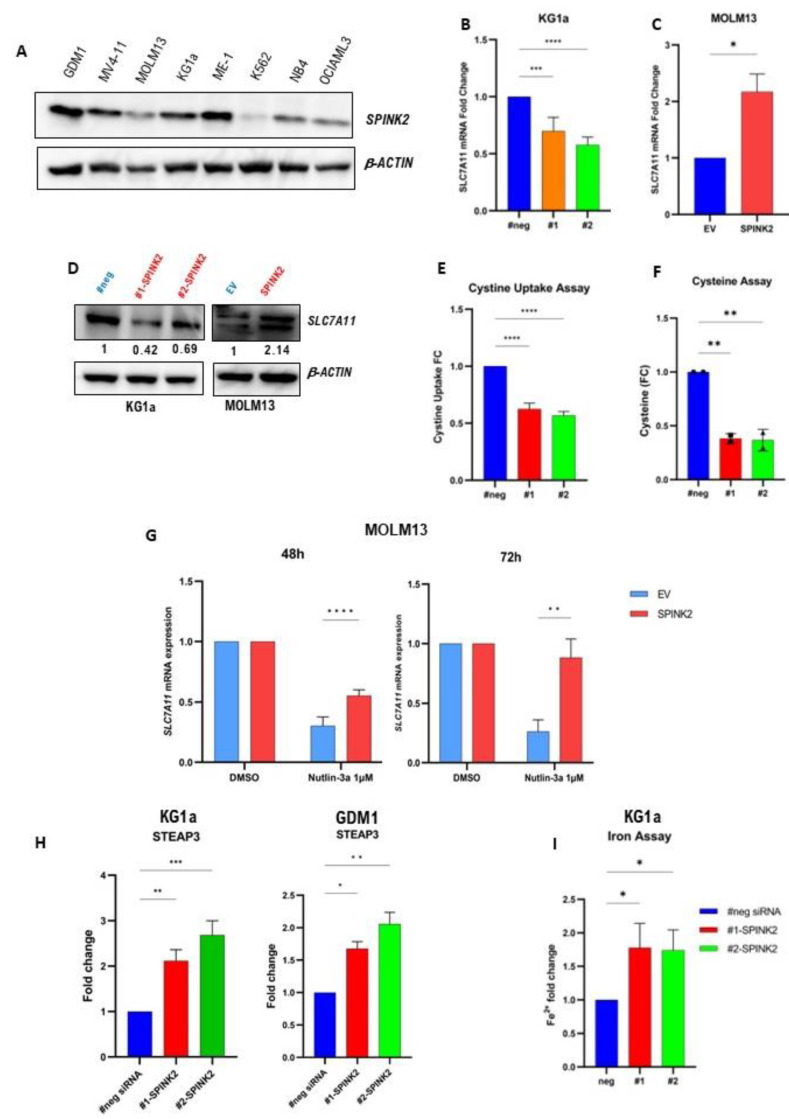
Transcriptome analysis reveals potential link between SPINK2 and ferroptosis-related genes. (**A**) Western blot showing the protein expression of *SPINK2* in various AML cell lines. *β-ACTIN* was used as loading control. (**B**) *SLC7A11* mRNA expression by qPCR in KG1a negative-control siRNA (#neg) and *SPINK2* knockdown with 2 siRNAs (#1 and #2). Statistics: one-way ANOVA with Dunnett’s multiple comparisons test. Mean ± SD of four independent experiments is shown. (**C**) *SLC7A11* mRNA expression by qPCR in MOLM13 empty vector (EV) and *SPINK2*-overexpressing cells. Statistics: unpaired *t*-test. Mean ± SD of two independent experiments is shown. (**D**) Western blot showing *SLC7A11* expression upon *SPINK2* knockdown in KG1a cells and *SPINK2* overexpression in MOLM13 cells. The numbers denote the relative protein expression normalized to the loading control. *β-ACTIN* was used as loading control. (**E**) Cystine uptake assay in KG1a cells. Cystine uptake in SPINK2-KD cells vs. negative control, in which cystine uptake in negative control was set to a value of 1. Statistics: one-way ANOVA with Dunnett’s multiple comparisons test. Mean ± SD of three independent experiments is shown. (**F**) Intracellular cysteine assay in KG1a cells comparing cysteine levels in SPINK2-KD cells vs. negative control; cysteine level in negative control was set to a value of 1. Statistics: one-way ANOVA with Dunnett’s multiple comparisons test. Mean ± SD of two independent experiments is shown. (**G**) mRNA expression by qPCR of *SLC7A11* in MOLM13 cells treated for 48 h and 72 h with Nutlin-3a. Statistics: one-way ANOVA with Dunnett’s multiple comparisons test. Mean ± SD of two independent experiments is shown. (**H**) *STEAP3* mRNA expression by qPCR in KG1a and GDM1 cells with SPINK2-KD vs. negative control. Statistics: one-way ANOVA with Dunnett’s multiple comparisons test. Mean ± SD of three independent experiments in KG1a; mean ± SD of two independent experiments in GDM1. (**I**) Iron (Fe^2+^) assay in KG1a cells with *SPINK2*-KD vs. negative control; Fe^2+^ level in negative control was set to a value of 1. Statistics: one-way ANOVA with Dunnett’s multiple comparisons test. Mean ± SD of two independent experiments is shown. For all images: * *p* < 0.05, ** *p* < 0.01, *** *p* < 0.001, **** *p* < 0.0001.

**Figure 4 ijms-24-09696-f004:**
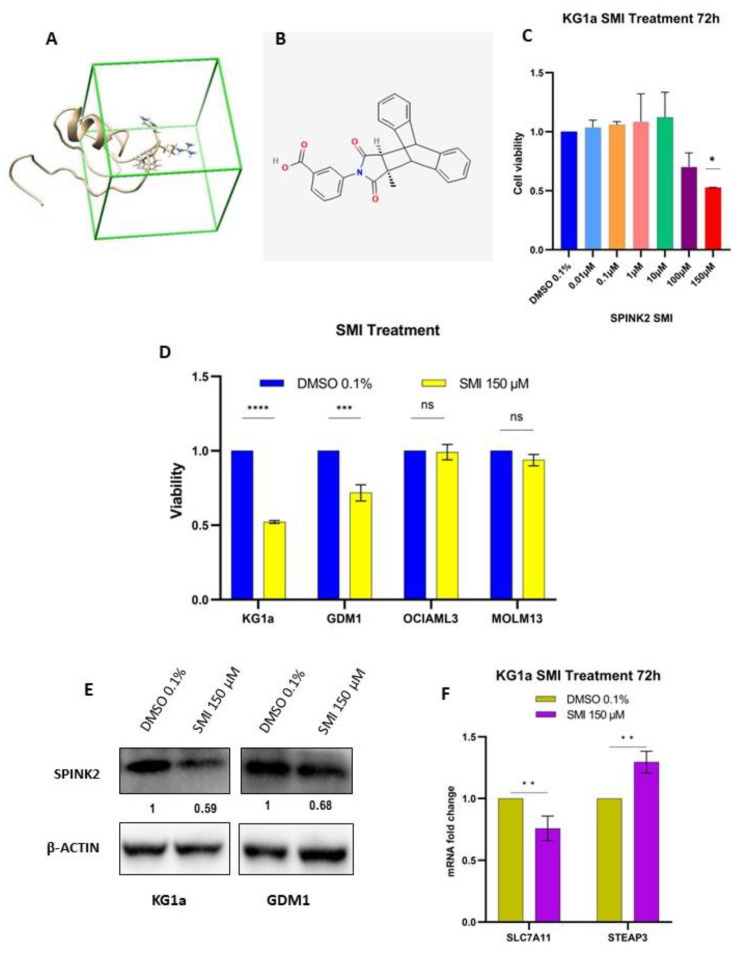
Identification and in-vitro testing of potential SPINK2 small-molecule inhibitor (SMI). (**A**) Targeting domain of SPINK2 protein highlighted by the green box. (**B**) Chemical structure of the potential *SPINK2* SMI, C_26_H_19_NO_4_. (**C**) Cell viability analysis of KG1a cells treated with DMSO 0.1% and increasing doses of SPINK2 SMI for 72 h. Statistics: Ordinary one-way ANOVA with Dunnett’s multiple comparison’s test. Mean ± SD shown for two independent experiments. * *p* < 0.05. (**D**) Cell viability analysis of KG1a, GDM1, OCIAML3 and MOLM13 cells treated with SMI for 72 h at 150 µM. (**E**) Western blot showing SPINK2 expression in KG1a and GDM1 cells after 72 h of SMI 150 µM treatment. The numbers denote the relative protein expression normalized to the loading control, β-ACTIN. (**F**) qPCR for *SLC7A11* and *STEAP3* in KG1a cells treated for 72 h with DMSO 0.1% and SMI 150 µM. Statistics: one-way ANOVA with Tukey’s multiple comparisons test. Mean ± SD is shown for three independent experiments. **** *p* < 0.0001, *** *p* < 0.001,** *p* < 0.01, * *p* < 0.05, ns, not significant.

**Figure 5 ijms-24-09696-f005:**
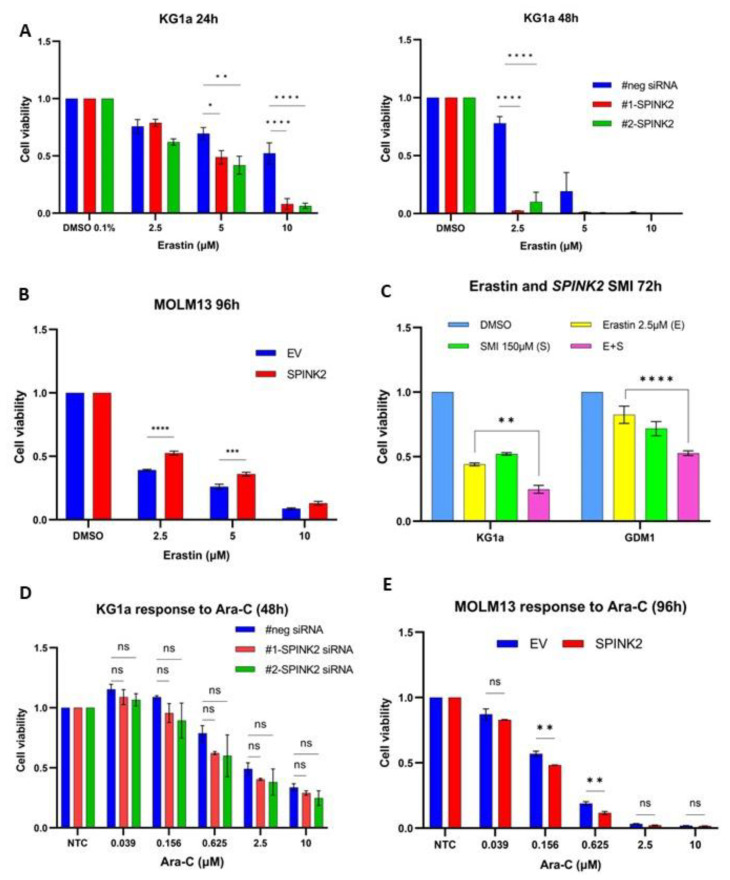
Effects of modulation of SPINK2 expression on erastin sensitivity. (**A**) Cell viability analysis of KG1a cells with siRNA knockdown (#neg, #1-SPINK2, #2-SPINK2) treated with DMSO and erastin for 24 h (**left**) and 48 h (**right**) at the indicated doses. (**B**) Cell viability analysis of MOLM13 cells (EV and SPINK2) treated with DMSO and erastin for 96 h at the indicated doses. (**C**) Cell viability analysis of KG1a wildtype and GDM1 wildtype cells treated with DMSO, erastin and SPINK2-SMI for 72 h at the indicated doses. (**D**) Cell viability analysis of KG1a cells with siRNA knockdown (#neg, #1-SPINK2, #2-SPINK2) treated with Ara-C for 48 h at the indicated doses. (**E**) Cell viability analysis of MOLM13 (EV and SPINK2) cells treated with Ara-C for 96 h at the indicated doses. For all images: Statistics: one-way ANOVA with Tukey’s multiple comparisons test; Mean ± SD is shown for at least two independent experiments. ns, not significant, * *p* < 0.05, ** *p* < 0.01, *** *p* < 0.001, **** *p* < 0.0001.

**Figure 6 ijms-24-09696-f006:**
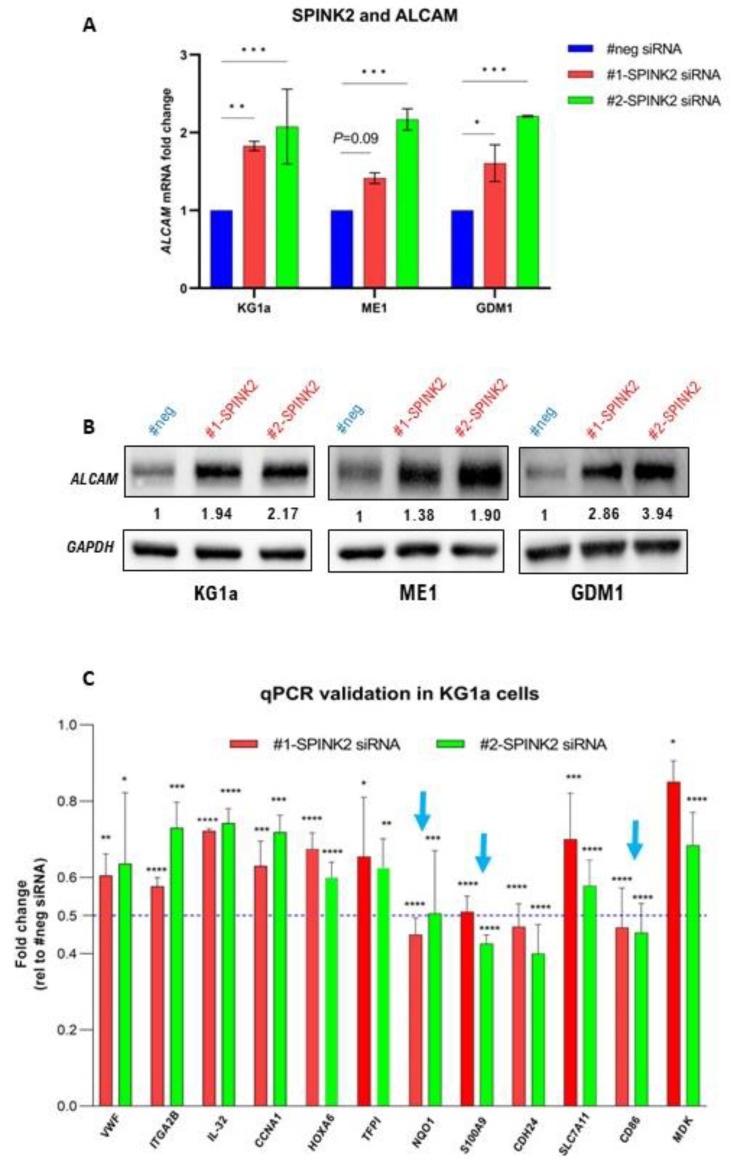
*SPINK2* knockdown modulates expression of immune-response related genes in LSC-like cells. (**A**) *ALCAM* mRNA expression by qPCR in KG1a, ME1 and GDM1 cells with negative control siRNA (#neg) and *SPINK2* knockdown with 2 siRNAs (#1 and #2). Statistics: one-way ANOVA with Dunnett’s multiple comparison’s test. Mean ± SD is shown for two independent experiments. (**B**) Western blots showing *ALCAM* expression in KG1a, ME1 and GDM1 cells with negative control siRNA (#neg) and *SPINK2* knockdown with 2 siRNAs. The numbers denote the relative protein expression normalized to the loading control. *GAPDH* was used as loading control. (**C**) mRNA expression by qPCR of several genes consistently downregulated by *SPINK2* knockdown in KG1a cells. Blue arrows indicate T-cell activity inhibitory genes (*NQO1, CD86, S100A9*) which were downregulated ≥ two-fold. Statistics: one-way ANOVA with Dunnett’s multiple comparisons test. Mean ± SD is shown for three independent experiments. For all images: * *p* < 0.05, ** *p* < 0.01, *** *p* < 0.001, **** *p* < 0.0001.

**Table 1 ijms-24-09696-t001:** Clinicopathological and mutational associations of SPINK2 protein expression in the PWH AML cohort.

Characteristic	High SPINK2 (*n* = 77)	Low SPINK2 (*n* = 95)	*p*-Value
**Sex**			
*Male*	41 (53.3%)	48 (50.5%)	0.76
*Female*	36 (46.7%)	47 (49.5%)	
**Age, years**			
*Median (range)*	54 (20–75)	51 (18–86)	0.23
**Hb level, g/dL**			
*Median (range)*	8.5 (3–13.6)	7.8 (2.9–12.9)	0.16
**Bone Marrow blast, %**			
*Median (range)*	69 (11–98)	69 (12–98)	0.65
**WBC level, ×10^9^/L**			
*Median (range)*	28.1 (1.3–517)	19.8 (0.9–330.4)	0.18
**Platelets, ×10^9^/L**			
*Median (range)*	67 (4–748)	43 (2–247)	<0.001
**FAB classification**			
*M0*	1/48 (2.1%)	4/68 (5.9%)	0.40
*M1*	11/48 (22.9%)	21/68 (30.9%)	0.40
*M2*	8/48 (16.7%)	17/68 (25.0%)	0.36
*M4 (incl. M4Eo)*	11/48 (22.9%)	13/68 (19.1%)	0.65
*M5*	16/48 (33.3%)	11/68 (16.2%)	0.04
*M6*	1/48 (2.1%)	2/68 (2.9%)	0.99
*Unclassified*	29/77 (37.6%)	27/95 (28.4%)	-
**AML type**			
*De novo*	71 (92.2%)	85 (89.5%)	0.61
*Secondary/t-AML*	6 (7.8%)	10 (10.5%)	
**MRC Cytogenetic Risk**			
*Favorable*	5 (6.5%)	19 (20.7%)	0.014
*Intermediate*	64 (83.1%)	61 (66.3%)	0.014
*Adverse*	8 (10.4%)	12 (13.0%)	0.64
*Unclassified*	-	3	-
**ELN 2022 risk**			
*Favorable*	19/73 (26.0%)	39/92 (42.4%)	0.033
*Intermediate*	34/73 (46.6%)	24/92 (26.1%)	0.009
*Adverse*	20/73 (27.4%)	29/92 (31.5%)	0.61
**Cytogenetics**			
*Normal*	51 (66.2%)	44 (47.8%)	0.019
*t(8;21)*	0 (0.0%)	14 (15.2%)	<0.001
*inv(16)*	5 (6.5%)	4 (4.4%)	0.73
*Complex*	5 (6.5%)	5 (5.4%)	0.99
*Others*	12 (15.6%)	20 (21.7%)	0.33
*Unknown*	-	3 (3.2%)	-
**Mutations**			
*FLT3-ITD*	26/77 (33.8%)	21/95 (22.1%)	0.12
*NPM1*	33/77 (42.9%)	14/95 (14.7%)	<0.0001
*CEBPA bZIP*	2/74 (2.7%)	18/95 (19.0%)	0.001
*DNMT3A*	26/74 (35.1%)	18/95 (19.0%)	0.022
*NPM1+/DNMT3A+*	17/74 (23.0%)	7/95 (7.4%)	0.007
*NPM1+/FLT3-ITD+*	20/77 (26.0%)	11/95 (11.6%)	0.017
*NPM1+/FLT3-ITD+/DNMT3A+*	9/74 (12.2%)	6/95 (6.3%)	0.275
*TP53*	1/69 (1.5%)	2/86 (2.3%)	0.99
*RUNX1*	8/69 (11.6%)	12/86 (14.0%)	0.81
*ASXL1*	4/69 (5.8%)	4/86 (4.7%)	0.99
*BCOR*	2/69 (2.9%)	2/86 (2.3%)	0.99
*EZH2*	1/69 (1.5%)	3/86 (3.5%)	0.63
*SF3B1*	0/69 (0.0%)	1/86 (1.2%)	0.99
*SRSF2*	4/69 (5.8%)	4/86 (4.7%)	0.99
*STAG2*	1/69 (1.5%)	6/86 (7.0%)	0.13
*U2AF1*	0/69 (0.0%)	2/86 (2.3%)	0.50
*ZRSR2*	2/69 (2.9%)	0/86 (0.0%)	0.20

Hb, hemoglobin; WBC, white blood cell count; FAB, French–American–British Classification; MRC, Medical Research Council; ELN, European LeukemiaNet; ITD, internal tandem duplication; bZIP, basic-region leucine zipper motif.

**Table 2 ijms-24-09696-t002:** Associations of SPINK2 expression with therapy outcomes in AML.

Factor	High SPINK2	Low SPINK2	*p*-Value
Whole cohort (N = 137)	N = 60	N = 77	
**Response to induction**			
CR	73.3%	88.3%	0.028
NR1	51.7%	33.8%	0.038
**Relapse ^‡^ after CR**			
Median RFS	9 months	37 months	0.004
6-month relapse rate	31.8%	9.1%	
5 year RFS	25.8%	46.8%	
*Intermediate cytogenetic risk (N = 101)*	*N = 51*	*N = 50*	
**Response to induction**			
CR	68.6%	90.0%	0.01
NR1	66.7%	37.5%	0.005
**Relapse ^‡^ after CR**			
Median RFS	12 months	37 months	0.018
6-month relapse rate	31.4%	6.9%	
5 year RFS	27.0%	44.6%	
*Intermediate risk ELN2022 (N = 47)*	*N = 28*	*N = 19*	
**Response to induction**			
CR	67.9%	84.2%	0.31
NR1	67.9%	21.1%	0.003
**Relapse ^‡^ after CR**			
Median RFS	14 months	37 months	0.034
6-month relapse rate	26.3%	6.7%	
5 year RFS	17.9%	34.3%	
*Normal karyotype (N = 76)*	N = 40	N = 36	
**Response to induction**			
CR	72.5%	91.7%	0.040
NR1	55.0%	27.8%	0.021
**Relapse** ** ^‡^ ** **after CR**			
Median RFS	12 months	35 months	0.07
6-month relapse rate	31.0%	6.2%	
5 year RFS	30.2%	41.9%	
*NPM1^mut^ (N = 46)*	N = 33	N = 13	
**Response to induction**			
CR	75.8%	100%	0.08
NR1	48.5%	15.4%	0.049
**Relapse** ** ^‡^ ** **after CR**			
Median RFS	14 months	Unreached	0.095
6-month relapse rate	28.0%	0.0%	
5 year RFS	35.5%	50.5%	

**^‡^** Relapse rates were calculated only for patients who achieved CR. Abbreviations: CR, complete response achieved, irrespective of number of inductions; NR1, non-response at first induction; RFS: relapse-free survival.

**Table 3 ijms-24-09696-t003:** Multivariate analysis for OS, EFS and RFS.

Covariates	OS			EFS			RFS ^§^		
Whole Cohort ^‡^ N = 125 ^¶^	HR	95%C.I.	*p*-value	HR	95%C.I.	*p*-value	HR	95%C.I.	*p*-value
*Age ≥ 60 years*	1.29	0.68–2.36	0.416	/	/	/	/	/	/
*SPINK2^high^*	2.45	1.48–4.07	<0.001	2.08	1.31–3.32	0.002	1.89	1.12–3.15	0.015
*CR1*	0.40	0.24–0.67	<0.001	0.33	0.21–0.52	<0.001	/	/	/
*SCT ^¥^ in CR*	0.11	0.02–0.37	0.0023	0.15	0.04–0.36	<0.001	0.11	0.02–0.37	0.003
*DNMT3A*	1.20	0.72–1.96	0.479	1.18	0.73–1.87	0.490	1.602	0.91–2.73	0.090
*ELN2022 adv*	1.78	1.02–3.02	0.037	1.86	0.94–3.43	0.060	2.16	1.21–3.73	0.007
*IDH2*	2.33	1.18–4.31	0.010	1.58	0.93–2.59	0.080	/	/	/
*NPM1^mut^* ^†^ N = 42 ^¶^	HR	95%C.I.	*p*-value	HR	95%C.I.	*p*-value	HR	95%C.I.	*p*-value
*Age ≥ 60 years*	9.10	2.36–34.39	0.001	7.53	2.01–27.45	0.002	3.58	0.92–12.13	0.046
*SPINK2^high^*	5.55	1.89–21.32	0.005	5.11	1.91–16.65	0.003	3.52	1.23–11.72	0.027
*FLT3-ITD*	2.54	0.94–8.18	0.085	3.9	1.37–11.94	0.017	2.47	0.88–7.84	0.100
*DNMT3A*	0.81	0.34–1.99	0.635	1.10	0.49–2.57	0.824	3.12	1.20–9.65	0.029

CR1: Complete remission at first induction; SCT in CR: stem cell transplantation administered after achieving complete remission; ELN2022 adv: ELN2022 adverse risk; ITD: internal tandem duplication; HR: hazard ratio; C.I.: confidence interval. ^§^ For RFS analysis, only patients eventually achieving CR were included in the analysis in all cohorts (whole, N = 108; *NPM1^mut^*, N = 38). ^‡^ The covariates included in the multivariate analyses are those which demonstrated significant associations (*p* < 0.05) in univariate survival analyses (Appendix AA,B). ^¥^ These only include patients from the PWH SCT cohort (N = 37). ^†^ The covariates included in NPM1 analysis are those which are part of ELN2022’s criteria (*FLT3*-ITD) and generally associated with poor prognosis in *NPM1^mut^* patients (age, *DNMT3A*). ^¶^ Patients were only included if they had complete cytogenetic and mutational data which allowed them to be assigned to an ELN 2022 risk category.

## Data Availability

For data sharing of the results generated in the study, please contact the corresponding author at margaretng@cuhk.edu.hk. Publicly available datasets were also analyzed in this study, which are accessible through the NCBI GEO database using the following accession numbers: GSE7186, GSE13164, GSE13159, GSE995, GSE30029, GSE30377, GSE6891, GSE17855 and GSE24006. Other databases such as cBioPortal (https://www.cbioportal.org, accessed on 25 July 2020) and the TCGA portal (https://portal.gdc.cancer.gov/, accessed on 14 February 2020) were also accessed for data.

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
