# Peer review of "SPINK2 Protein Expression Is an Independent Adverse Prognostic Marker in AML and Is Potentially Implicated in the Regulation of Ferroptosis and Immune Response"

_ijms, 2023, doi:10.3390/ijms24119696_

Round 1

Reviewer 1 Report

Summary: In the present article, the authors assessed the protein expression, clinicopathological, and prognostic significance of serine protease inhibitor kazal type 2 (SPINK2) in AML, and examined its potential biological functions. High SPINK2 protein expression was an independent adverse biomarker for survival, and an indicator of elevated therapy resistance and relapse risk. RNA sequencing analyses revealed a potential role for SPINK2 in ferroptosis and immune regulation. Finally, they identified a putative small molecule inhibitor for SPINK2 that is worthy of future investigation. The article is well written, and I have only minor comments for improvement. Please see below:

1)      Lines 56-61: This entire paragraph belongs in the Results section, not in the Introduction. Please revise accordingly.

2)      The Introduction could be improved by adding a clearly stated hypothesis explaining the rationale for the study.

3)      The authors identified SPINK2 through a LSC-associated gene signature. However, the expression of SPINK2 in AML stem versus progenitor cells was not mentioned. Is SPINK2 more highly expressed in quiescent LSCs compared with proliferating progenitor cells?

4)      SPINK2 has already been published as a poor prognostic indicated in AML: PMIDs 31452767, 33925480, 35173838, 36865386. These articles were cited in the Discussion section.

Reviewer 2 Report

The article “SPINK2 protein expression is an independent adverse prognostic marker in AML, and is potentially implicated in the regulation of ferroptosis and immune response.” by Pitts HA, et al. reposted that high level of SPINK2 was a poor prognostic factor for AML in several dataset and related with some well-known patient characteristics, such as ELN2022 risk, FLT3-ITD, and so on. Additionally, the authors analyzed functions of SPINK2 in AML using several biochemistry methods, and anti-AML activity of a SPINK2 inhibitor. I considered that this article was very interesting, and suitable for publication for IJMS. However, I suggested several points for the purpose to improving your article as below.

1.       Using several datasets, high SPINK2 level predicted short survival time. Were the AML patients in these datasets included for survival time analysis when they received 3+7 conventional induction therapy? For instance, were the patients treated with high-dose cytarabine excluded? The author should add the inclusion criteria in “Methods”.

2.       Allogenic hematopoietic stem cell transplantation overcame the negative impact of high SPINK2 level in overall population, but did not in the patients with relapse after CR1 and refractoriness, suggesting that allogenic SCT could improve clinical outcome only for CR1 patients. Why did the author consider about these results happened? The authors could add several comments in “Discussion”.

3.       Currently, venetoclax is a key drug for AML. Is venetoclax effective for AML with high SPINK2 level? The authors could add several comments if possible from biochemical and clinical points of view.
